# Evaluation of the Street Canyon Level Air Pollution Distribution Pattern in a Typical City Block in Baoding, China

**DOI:** 10.3390/ijerph191610432

**Published:** 2022-08-22

**Authors:** Jingcheng Zhou, Junfeng Liu, Songlin Xiang, Yizhou Zhang, Yuqing Wang, Wendong Ge, Jianying Hu, Yi Wan, Xuejun Wang, Ying Liu, Jianmin Ma, Xilong Wang, Shu Tao

**Affiliations:** 1Laboratory for Earth Surface Processes, College of Urban and Environmental Sciences, Peking University, Beijing 100871, China; 2School of Statistics, University of International Business and Economics, Beijing 100029, China

**Keywords:** air pollution, street canyon, urban canopy, urban morphology, traffic emissions

## Abstract

Urban traffic pollution, which is strongly influenced by the complex urban morphology, has posed a great threat to human health. In this study, we performed a high-resolution simulation of traffic pollution in a typical city block in Baoding, China, based on the Parallelized Large-eddy simulation Model (PALM), to examine the distribution patterns of traffic-related pollutants and explore their relationship with urban morphology. Based on the model results, we conducted a multi-linear regression (MLR) analysis and found that the distribution of air pollutants inside the city block was dominated by both traffic emissions and urban morphology, which explained about 70% of the total variance in spatial distribution of air pollutants. Excluding the contribution of emissions, over 50% of the total variance can still be explained by the urban morphology. Among these urban morphological factors, the key factors determining the spatial distribution of air pollution are “Distance from the road” (DR), “Building Coverage Ratio” (BCR) and “Aspect Ratio” (H/W) of the street canyon. Specifically, urban areas with lower Aspect Ratio, lower BCR and larger DR are less affected by traffic pollution. Compiling these individual factors, we developed a complex Urban Morphology Pollution Index (UMPI). Each unit increase in UMPI is associated with a one percent increase of nearby traffic pollution contribution. This index can help urban planners to semi-quantitatively evaluate building groups which tend to trap or ventilate traffic pollution and thus help to reduce human exposure to street canyon level pollution through either traffic emission control or urban morphology amelioration.

## 1. Introduction

Over 60% of the Chinese population live in urban areas, where traffic emissions are a main source of air pollutants. Fine particulate matter (PM_2.5_) and ozone (O_3_) can lead to 650,000 and 70,000 premature deaths in urban areas each year, respectively [1,2]. These deaths are closely related to urban residents who live close to street canyons and experience chronic exposure to traffic pollution. Therefore, understanding the source, dispersion and distribution of the air pollutants in urban areas is the key to reducing the exposure of urban residents and protecting human health.

The shape and clustering of building groups (including the form of street canyons) play an important role in determining the air flow movements, pollution dispersion processes and accumulation of pollutants in urban areas. Deep street canyons have been found to be associated with the formation of vortices, which remarkably strengthen the accumulation of pollutants at the ground level [3]. Fu et al. [4] found that canyons with building heights over 40 m and a high asymmetric ratio lead to a significant increase in human exposure to traffic pollution. However, as for the pollution distribution inside building groups, there is still substantial uncertainty regarding the effects of building groups on the dispersion processes.

To assess the contributions of building group shapes to air pollution distribution, a set of parameters describing the urban morphology have been proposed. For example, Edussuriya et al. [5] identified the correlations between NO_x_, CO, PM_2.5_ and urban morphological indicators (e.g., the aspect ratio, the building compactness and mean building height) in Hong Kong and found that in-site fabrics have strong impacts on air quality. Yang et al. [6] revealed that the pedestrian wind speed ratio has good correlation with several parameters, including the building density and average height. However, most of the previous studies mainly focused on the correlation with a single factor and ignored the overall effects of the urban morphology.

To investigate the spatiotemporal distribution patterns of air pollutants within urban block areas, we apply the Parallelized Large-eddy simulation Model (PALM) for a simulation of a typical urban areas in Baoding, China. Large Eddy Simulation (LES) models have been successfully applied in urban areas. For example, Idrissi et al. [7] applied an LES model to study air pollution dispersion within an area of 600 m × 580 m with complex urban morphology. Xavier et al. [8] used LES models to compare the performance of air pollution simulations on different types of urban layouts. With the model results, we specifically focus on the investigation of correlations between the urban morphological parameters and the spatiotemporal distribution of air pollutants, based on which we then introduce a new prediction index for quick positioning of the areas highly affected by air pollution in urban blocks. This index, calculated with the characteristics of urban forms, is aimed at quick semi-quantitative estimation of urban air pollution distribution, and will help policy makers and urban planners to optimize the planning of urban building groups to reduce urban air pollution.

The model and its performance are described in Section 2. We then investigate their constraints with both traffic emissions and different urban morphology parameters in Section 3.1. Section 3.2 contains the introduction to our newly developed urban morphology index. Finally, conclusions are drawn in Section 4.

## 2. Method

### 2.1. Model Description

In this study, we simulate the air pollution dispersions inside urban street canyons based on the Parallelized Large-eddy simulation Model (PALM) [9]. PALM has been used for a variety of boundary layer studies over the last 15 years, such as heterogeneously heated convective boundary layers [10,11], urban canopy flows [12,13] and cloudy boundary layers [14,15]. The participation in the first intercomparison of LES models in terms of the stable boundary layer also proved its ability to perform simulations with a resolution of down to 1 m [16].

PALM uses the governing equations of non-hydrostatic, filtered, incompressible Navier–Stokes equations in the Boussinesq-approximated form. An upwind-biased fifth-order differencing scheme [17,18] and a third-order Runge–Kutta scheme [19,20,21,22] were chosen for time and space discretization, respectively. More details about the PALM governing equations and the model structure can be found in Maronga et al. [23].

In this study, the PALM model is applied for the ultra-fine-solution, urban-scale air pollution simulation. PALM allows for a scalability of up to 50,000 processor cores, which allows for simulations with ultra-fine grids. Several previous studies have applied the PALM model for high-resolution simulations in urban atmospheric environments. With the newly developed PALM-4U (short for PALM for urban atmospheric boundary layers) components, the PALM model is especially suitable for the simulation of complex urban layouts, just as we investigate in this study.

### 2.2. Model Configuration and Evaluation

#### Study Area and Time

A 1 km × 1 km × 200 m city block in the downtown area of Baoding, China, was chosen as the study area (see Figure 1). Baoding city is located about 200 km south of Beijing. It is an important source of air pollution over the Beijing–Tianjin–Hebei (BTH) region and also suffers from severe air pollution problems itself. Previous studies have shown that Baoding is the major contributor to air pollution over the entire BTH area [24,25]. Therefore, Baoding is a representative city for urban pollution studies. As shown in Figure 1, the study area consists of three main streets and lots of buildings with different heights (see Appendix A for the satellite map). We divided the study area into 200 × 200 × 40 grids, with the size of each grid equal to 5 m × 5 m × 5 m. The simulation period ranged from 22 July 2018 to 29 July 2018. The time resolution of the model output was 60 s.

### 2.3. Model Configuration and Initialization

The PALM model system, version 6.0, revision 4233, is used in this study. The default “clear-sky” scheme is used for calculating radiation fluxes. The clear-sky scheme is a simple model calculating the shortwave incoming, shortwave outgoing, longwave incoming, longwave outgoing and, consequently, the net radiation at the surface. The land surface model and the urban surface model of PALM are used, respectively, for natural-type surfaces (e.g., vegetation, soil) and building surfaces. The surface roughness length is set based on the results from the static driver.

A chemical mechanism based on the PHSTAT mechanism (abbreviation for photostationary), which is one of the mechanisms included in PALM-4U, is applied for chemistry simulation. The PHSTAT mechanism is a simplified two-reaction mechanism describing the photostationary equilibrium between NO, NO_2_ and O_3_. We add CO as a new species into the mechanism and treat it as a passive gas, since it hardly takes part in chemical reactions in this scene. Particulate matter (PM) is not included in this study, because PM generally consists of primary and secondary components. For primary components, they hardly undergo chemical processes, and therefore show a similar pattern to CO. For secondary components, the chemical mechanism in this study is not sufficient for simulation. As a result, we choose to focus on the chemical processes between NO_x_ and O_3_.

The boundary conditions (including the wind fields, chemical species and a list of other profiles, such as *θ* and *q*, were obtained from WRF-Chem (version 4.2) simulation results. The emission inventory used for WRF-Chem simulation is taken from the MEIC model. Due to the difference in spatial resolution between the two models, data from two national monitoring sites located to the northeast and southwest of the study area were used for calibration for different boundaries. The data retrieved from the northeast site were applied to set the north and east boundaries of the area, while the data acquired from the southwest site were used to establish the south and west boundaries. The top boundary was set based on the mean of the data obtained from the two sites. The position of the national sites and the study area are shown as Appendix A.

Previous studies have reported that, compared to traffic emissions, residential emissions are minor in urban areas [26,27,28,29,30,31]. Moreover, there are no industrial facilities in the study area. Therefore, only traffic emissions are included in the simulation. During this study period, we counted the vehicle numbers on different roads via camera footage. The vehicles were categorized into four types: light- and heavy-duty passenger vehicles and light- and heavy-duty trucks. The number of each type of vehicle was multiplied by a corresponding emission factor to estimate traffic emissions. The emission factors were obtained from Chen et al. [32].

### 2.4. Model Evaluation

We validated the model results using the data from the two nearest monitoring sites (see Appendix A for the position of the sites). Figure 2 shows the time series of the observed and modelled species, i.e., O_3_, NO_2_ and CO, from 22 July 2018 to 29 July 2018 (a scatter plot is also attached, please refer to Appendix A). Generally, the simulation results fit the observations well during most of the study time, and the results from monitoring site 2 are closer to the model results than those from monitoring site 1. This is because monitoring site 2 is located closer to the study area. For O_3_, both the model and observation data show similar diurnal cycles, which is related to the photochemistry of O_3_. Some exceedance in the morning can be explained by the morning traffic peak, which contributes to large amounts of NO_x_ emissions. The modelling results for CO attain the best fit with the monitored data, which is because the lifetime of CO is much longer than that of NO_2_, and it is less influenced by chemical reactions. In addition, we also measured the CO and NO_2_ concentrations at a roadside site during the morning and afternoon from 27–29 July (Appendix A). The model results show relatively good fits with the roadside monitoring data. The results of NO_2_ are slightly higher than the monitored data, while the results of CO are generally in agreement. Another simulation on an area of the same size (we call it the “test area”), where the national site 2 is located, was also conducted. (Appendix A) The results show that PALM can reach satisfying simulation results. Overall, the model results show high consistency with the observational data and are reliable for further analysis.

### 2.5. Data Analysis

#### Parameters Describing the Urban Form

The interpretation of urban form in different studies can be quite different. Previous studies regarding urban morphology are mainly based on qualitative descriptions of compactness and building heights [33]. Here, we adopted some parameters from Adolphe et al. [34] to describe the morphological characteristics of the urban block area, including the building coverage ratio (BCR), rugosity, porosity and occlusivity more precisely. In addition, two important parameters describing the street canyon form were taken into consideration: the asymmetry ratio (*H*_1_/*H*_2_) and aspect ratio (*H*/*W*) [35]. The corresponding definitions of these parameters and their calculations are shown in Table 1 (for detailed definitions of the symbols in the equation, please refer to Appendix A).

Some of these parameters are easy to understand, while others are not. For example, distance from the main road (DR), BCR and Rugosity are generally clear indices, which, respectively, describe a certain grid’s position, along with the density and height of its surrounding area. For other parameters which are not so straightforward, we give a brief discussion here. There are several different ways of interpreting the concept of porosity. Here, we adopt the definition given by T. Gál et al. [36], which is an index for measuring how penetrable the area is for the airflow. Therefore, areas with high porosity are generally associated with low building density (BCR), but there still remain some exceptions. Occlusivity is another important index, which, to some extent, provides 3-D information of the area. It is defined as the average of the ratios of the perimeter of built area to unbuilt area on all the layers. The layer height here is chosen to be 3 m, which is generally the height of a storey. Note that a building’s margin is calculated both in the perimeter of built and unbuilt areas, so a high occupation rate of buildings can lead to a high ratio in its layer. The difference between occlusivity and the BCR lies in that low buildings have relatively lower contributions to occlusivity, while they are completely calculated in the BCR. In high layers where few buildings exist, the low ratio of the perimeters can significantly lower the occlusivity. Therefore, to distinguish from BCR, occlusivity is defined as the “openness” of the urban areas. As for the street canyons, aspect ratio and asymmetry ratio are commonly used indices. The only thing to be noted is that the asymmetry ratio is between 0 and 1 in our study. A multicollinearity test is attached in the Appendix A to confirm that no severe collinearity exists between the parameters (Appendix A).

### 2.6. Multi-Linear Regression

The parameters described in Table 1, along with the emission values on the road areas, were used to build up a multi-linear regression model. Two parameters, i.e., asymmetry ratio and the aspect ratio, are related to street canyon characteristics and are calculated based on the topography data in the range of −50 to +50 m along the direction of the road. The other urban morphological parameters were calculated based on the building groups in an area of 100 m × 100 m, with the grid located at the centre. If the grid was not on the main street, the distance to the nearest main street was also adopted. In this way, each grid contained a group of parameters representing the form of its neighbouring buildings. The spatial patterns of these parameters are shown in Appendix A. These parameters serve as the independent variables in the regression models. Since all the parameters are from or calculated from measured data, the variables in the regression model are all continuous data.

## 3. Results and Discussion

### 3.1. Effects of Urban Morphological Factors on Dispersion of Traffic Pollutants

The spatial distribution of the air pollutants can be found in Appendix A in the SI. An EOF analysis is also conducted to determine the contribution of various factors on the studied area. From the results, ground-level distribution of air pollution is determined not only by traffic-related emissions, but also by other factors, including background transport and urban morphology (please refer to Appendix A for detailed discussion). To quantitatively understand the contribution from individual factors, we constructed a multi-linear regression model and used traffic emissions and the abovementioned seven urban morphological parameters as the independent variables to explore their association to the mean concentration distribution of individual air pollutants. For each grid at the ground level over the study domain, we have a group of data, including mean pollution concentration (Appendix A), emissions and seven building morphological parameters (Appendix A). The grids occupied by buildings are omitted and we finally obtain 23,338 sets of data. The regression results are shown in Table 2 (see Appendix A for the standard error). The algebraic equation of the regression model in Table 2 takes the following form:(1)$$C=\beta_{0}+\beta_{1}*E+\beta_{2}*\mathrm{DR}+\beta_{3}*\left(H/W\right)+\beta_{4}*\left(H_{1}/H_{2}\right)+\beta_{5}*\mathrm{BCR}+\beta_{6}*\mathrm{Rugosity}\;+{\;\beta }_{7}*\mathrm{Porosity}\;+\beta_{8}*\mathrm{Occlusivity}+\epsilon$$
where C refers to concentration and E refers to emission. Other abbreviations can be found in Table 1. The unit for emission is “µg·m^−2^·s^−1^”, and units for DR and Rugosity are meters (m) Other variables are ratios and do not have units.

From Table 2, all pollutants except O_3_ have an R^2^ over 70%, showing that the selected eight parameters can explain most of the spatial distribution of traffic pollutants. O_3_ has an R^2^ of 47%, much lower than the other pollutants, indicating that O_3_, as a secondary pollutant, is only indirectly related to traffic emissions (through the NO_x_ titration process). These traffic-related parameters, e.g., emissions and DR, are generally having opposite effects on O_3_ compared to other pollutants.

For NO_2_ and CO, the correlation for all parameters is generally significant, except for the correlation between “Rugosity” and CO. Among the eight parameters, “DR”, “Rugosity” and “Occlusivity” are negatively correlated with the spatial distribution of air pollution. The negative correlation of “DR” shows that the longer the distance from the road, the lower the ground-level pollution concentration, mainly reflecting the dispersion pattern of traffic emissions. The coefficient of NO_2_ is about twice that of CO, showing that NO_2_ has a larger change when going away from the main roads. Increased building heights can lead to deeper canyons, where the ground-level wind speed becomes larger and contributes to a quicker dispersion, resulting in lower pollution levels. This is why “Rugosity” has a negative correlation. As previously discussed, “Occlusivity” indicates the openness of the building groups, and the negative correlation with pollution distribution indicates that areas with higher openness have higher pollution levels. However, by common sense, higher “Occlusivity”, i.e., lower openness, should give rise to worse ventilation and, finally, lead to higher pollution. Therefore, to explain this contradiction and reveal the true effect of “Occlusivity”, we also conducted an additional correlation analysis to differentiate the effects between road and non-road areas (Appendix A). The results show that “Occlusivity” is the only parameter with opposite correlation results between road and non-road areas. Over the non-road areas, “Occlusivity” has a positive correlation, just as expected. The negative correlation in road areas, however, is because the lower openness in the roads is related to narrow parts of the streets, where traffic numbers and its related emissions are lower. Since traffic emissions are very important in the studied block, the correlations with road areas are more significant in the correlation, and the combination of the different effects over road and non-road areas finally results in the negative correlation.

The remaining five parameters are all positively correlated with pollution distribution. “Emission” is the source of pollutions, and this result is expected. As for “Aspect Ratio” and “Asymmetry”, which describe the shape of street canyons, higher “Aspect Ratio” (which implies a deeper street canyon) and “Asymmetry” values (in this case closer to 1, which implies a more asymmetric canyon) lead to a higher pollution concentration within the street canyon, which is consistent with previous studies. “BCR” is related to building density, and areas with higher “BCR” values usually have worse ventilation, resulting in ground-level pollution accumulation. “Porosity” reflects the open volume ratios of the building groups. Higher “Porosity” values mean there is less space occupied by buildings, which can enhance the air flow and lead to stronger ventilation. However, another effect, which is that traffic-emitted pollutants are carried by the air flow into the residential areas, also have significant influence and override the effects of dispersion in this case, and finally results in the positive correlation.

The correlation patterns for O_3_ are generally opposite to the results of NO_2_ (Appendix A), as the titration by NO_x_ consumes O_3_ and increases NO_2_ concentrations at the same time. Therefore, in areas where NO_2_ have high concentration, O_3_ has relatively low concentration. This also indicates that controlling traffic emissions over a small city block may lower local NO_x_ and CO concentrations, but has little effect on O_3_. The most effective way to control O_3_ concentration over the urban area is to reduce the background contribution, which requires sophistically controlling both VOCs and NO_x_ emissions over a much broader area.

As previously discussed, traffic emissions are the most important factor in the area. Therefore, it accounts for a large proportion of the pollution distribution. Nevertheless, the effects of urban form also have important effects. To further explore the effects of city form on pollution distribution, we also conducted a correlation with the emission term excluded. The results are shown in Table 3 (see Appendix A for the standard error). The algebraic equation of the regression model in Table 3 takes the following form:(2)$$C=\beta_{0}+\beta_{1}*\mathrm{DR}+\beta_{2}*\left(H/W\right)+\beta_{3}*\left(H_{1}/H_{2}\right)+\beta_{4}*\mathrm{BCR}+\beta_{5}*\mathrm{Rugosity}\;+\beta_{6}*\mathrm{Porosity}\;+{\;\beta }_{7}*\mathrm{Occlusivity}+\epsilon$$
where C refers to concentration. Other abbreviations can be found in Table 1. The units for DR and Rugosity are meters (m). Other variables are ratios and do not have units.

Without the effects of emission, the regression models for NO_2_ and CO still have an R^2^ over 50%, showing that urban form still has reasonably large effects on pollution distribution. Compared to the previous regression, the results for individual morphology parameters are similar, except that “Rugosity” becomes insignificant in this case. These results indicate that a better designed urban form may be helpful for enhancing air ventilation and reducing pollution accumulation, resulting in a better urban air quality and a lower human exposure with the same traffic emissions.

We also calculated the association rates between the dependent and each independent variable in both road areas and non-road areas. All the independent and dependent variables are normalized into the range of 0 to 1. The simple correlation results are shown in Table 4 (see Appendix A for the standard error). The regression coefficient of each independent variable here means the rate of the change of the dependent variable is associated to one unit change in the corresponding independent variable. For example, the regression coefficient between the standardized NO_2_ concentration and the standardized NO_2_ emission over the road area is 0.46, which means that every additional percentage of emission leads to a 0.46% increase in NO_2_ concentration.

The results show that the correlation coefficients over the road areas are generally larger than those in the non-road areas. This is mainly because the emissions in the road areas can lead to larger rangeability in the concentration than in the non-road areas. From the results, “Emission”, “Aspect Ratio”, “Asymmetry” and “Rugosity” have major impacts, i.e., an additional increase/decrease of over 10% occurs when these factors have a 1% change. Therefore, the above indices should be given more attention in future urban planning. Another interesting finding is that the correlation coefficient between O_3_ and “Rugosity” (which is 0.73), indicating that high buildings should be avoided alongside the streets, as they cause large O_3_ accumulation in the street canyons. As for the non-road areas, the “DR” is much larger than other parameters, which indicates that the distance to the pollution source is the dominant factor over the non-road areas. The other parameters, e.g., BCR, Rugosity, Occlusivity and Porosity, though associated with smaller coefficients compared to “DR”, can also significantly affect the traffic pollution dispersion, and thus could serve as the fine-tuning parameters in urban canopy design.

### 3.2. Urban Morphology Pollution Index (UMPI)

In order to collectively address how the building morphology characteristics affect pollution dispersion without the influence from variation of traffic emission, we developed an Urban Morphology Pollution Index (UMPI, Equation (1)) based on the multiple regression coefficients between the standardized traffic-related pollution concentrations and urban morphological factors (Appendix A). The final coefficient in the equation is obtained using the average of the coefficients for NO_2_ and CO in Appendix A. This is because O_3_ has almost opposite correlation results to NO_2_, and it is mainly affected by the background concentration instead of morphology. The coefficient with the emissions is excluded and only the coefficients with morphological parameters remain to make sure that this index is only affected by the shape of the urban canopy. A rescale of the coefficients is then performed so that that every unit change of UMPI is associated with a 1% change (increase/decrease) of the corresponding pollutant. The final equation of the UMPI is as follows:(3)$$\mathrm{UMPI}=-25*p\left(\mathrm{DR}\right)+14*p\left(H/W\right)+22*p\left(H_{1}/H_{2}\right)+18*p\left(\mathrm{BCR}\right)\;-5*p\left(\mathrm{Rugosity}\right)-7*p\left(\mathrm{Occlusivity}\right)+5*p\left(\mathrm{Porosity}\right)+30$$
where p(*X*) represents each normalized urban morphological parameter (Equation (2)).
(4)$$P\left(X\right)=\frac{X-X_{min}}{X_{max}-X_{min}}$$
where *X* represents the original value of each parameter at a specific grid; *X_min_* is the minimum value of *X* over the whole domain; and *X_max_* is the maximum value of *X* over the whole domain.

Figure 3 shows the spatial distribution of the UMPI of each grid over the 1 km × 1 km study domain. The correlation between the UMPI and the normalized concentrations are shown in Table 5. Results show that for the studied pollutants (i.e., NO_2_, O_3_, CO), an additional 1% increase/decrease occurs when the UMPI changes by 1 unit. This shows that the UMPI can represent the general effects of building morphology on the air pollution and can be used for a quick pollution estimation.

The UMPI can be used to evaluate how the size and shape of building clusters potentially enhance traffic-related air pollution. Figure 4 marks three typical areas with different UMPIs in the study area, and further decomposes the contribution of individual morphological parameters to the UMPI. From these two figures, the Aspect Ratio and the BCR are the main reasons for the high UMPIs, while DR and the Rugosity mainly explain the lower values. Area ① is typical for the low UMPIs in residential areas. These areas are generally a reasonable distance from the roads, and thus have lower UMPIs. Area ② represents the high UMPIs in the road areas, which are mainly caused by the Aspect Ratio and the Porosity. In particular, the buildings next to area ② are higher (notice the difference in the grayscale of the building), and this leads to a higher Aspect Ratio, thus a higher UMPI. As for Area ③, even though it is close to the road, the high buildings have higher Rugosity, and contribute to a low UMPI.

We further collected the data from the “test area” for an additional test of the performance of the UMPI in other different areas. The results show that, except for NO_2_, the other two pollutants still keep the 1% relationship with the UMPI (Appendix A). The coefficient with NO_2_ is about 0.7%, which is much smaller than the previous 1%. This is because NO_2_ has the shortest lifetime and is mostly affected by local factors, e.g., traffic emissions and the dispersion conditions. Note that the UMPI is completely based on the characteristics of the urban canopy and is not related to traffic emission, meteorology or any other influencing factors. The similar relationship obtained in two different areas provides powerful support for the effectiveness of the UMPI.

The results in two different areas show that the effects of the urban morphological parameters are commonly shared. However, as the difference in the NO_2_ results shows, some adjustment may be necessary when using this index for quantitative estimation. Nevertheless, we believe that the UMPI can be widely used in the estimation of pollution conditions with higher efficiency. For example, there is no need for urban planners to carry out a thorough investigation over the whole city. Instead, they can use this index to position the pollution hotspots in urban areas and apply specific improvements over these areas to achieve a better urban design.

## 4. Conclusions

In this study, we adopted an LES model, namely, the PALM, to simulate the air flow and pollution distribution in a city block of Baoding, China, and applied an MLR model to identify the factors influencing the pollution distribution in the city block. The results reveal that traffic emissions are the most important factor affecting the concentration in the studied city block, which is in line with previous studies. Traffic emissions mainly occur within street canyons, and greatly affect the ground-level concentration in nearby areas. Nevertheless, the effects of traffic emissions can decrease quickly with an increase in the distance from the road. Therefore, to mitigate human exposure to traffic exhaust, residential areas should be located beyond a certain distance from roads, and higher-storey buildings are preferred to enhance ventilation.

Urban morphological forms (i.e., the shapes of street canyons and building groups) are also proved to be important factors influencing pollution distribution. In regard to street canyons, the “Aspect ratio” is the most important factor. Street canyons with high “Aspect ratios”, i.e., deep and narrow canyons, tend to accumulate more pollution, because wider street canyons can enhance air exchange and ventilation to reduce pollution concentration. In future urban planning, deep and narrow street canyons should be avoided. As for the shape of building groups, our results show that the main influencing factors are the “BCR”, “Rugosity”, “Occlusivity” and “Porosity”. Among these factors, “Rugosity” is the only factor which has a negative correlation. Higher BCR refers to high density building groups, which is unfavorable for ventilation and air exchange, and hence causes pollution accumulation. “Occlusivity” represents the openness of building groups. Higher “Occlusivity” is also related to dense building groups, which are unfavorable for ventilation and pollution dispersion. “Porosity”, the index for the open volume rate of building groups, is related to both the air exchange rate and the vulnerability of building groups (i.e., the chance that pollutants are carried into the building groups). In this case, the effects of background transport and traffic emissions are larger than the elimination from dispersion, leading to the result that higher porosity is related to higher pollution concentrations. Meanwhile, higher “Rugosity”, i.e., higher building groups, can lead to an increase in ground-level wind speed, thus enhancing ventilation and weakening pollution accumulation.

Based on these influencing factors, we develop a new index, i.e., the UMPI, to comprehensively describe the effects of urban morphology on pollution distribution. Our results show that the UMPI has good correlation with the change of pollution concentration. A unit change in the UMPI leads to approximately a 1% change in the pollution concentration. The characteristics of the UMPI are close to the effects discussed above, where areas have low BCR and large DR can lead to a low UMPI. In addition, high buildings over the non-road areas can also help to reduce the UMPI, which is possibly related to the enhanced turbulence near high buildings.

In conclusion, the distribution of air pollutants within a certain city area is controlled by several factors altogether. Within the urban area, traffic emissions are important sources, and lead to the road-centered distribution pattern. As a result, residential areas should be separated from main roads by a buffer area. Deep and narrow street canyons should also be avoided to prevent accumulation within the canyon. Furthermore, the form of building groups should also be considered. In most cases, high-density buildings are unfavorable for pollution dispersion. High buildings with low density can enhance ground-level ventilation by increasing wind speed, and thus help eliminate pollutants. Therefore, relatively separated high buildings may be a better choice for future cities.

We admit that all results from this study are based on the particular urban area in Baoding, China. Due to the limitation of data and potential changes in urban forms or traffic patterns, it is possible that the findings in this study may not be fully applicable for other places. Moreover, the study is based on the data during summertime. It is possible that in winter, due to various factors (e.g., less active photochemical reactions, heating, etc.), the pollution situation may be different. In the future, we will apply the framework of the UMPI proposed in this study to more cities to extensively test its effectiveness. We will also collect data of more pollutants in order to include other pollutants (e.g., benzopyrene, PM) into the model. Simulations in other seasons will also be carried out to explore the changes in patterns of different seasons.

## Figures and Tables

**Figure 1 ijerph-19-10432-f001:**
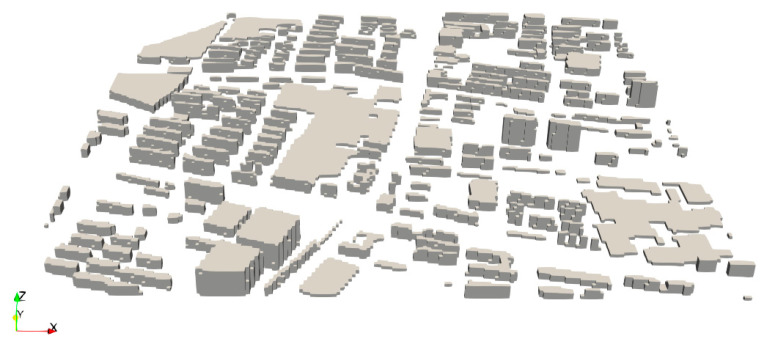
Three-dimensional view of the study area, a 1 km × 1 km × 200 m city block in the downtown area of Baoding, China. The grey cubes represent the shapes of the buildings within the study area.

**Figure 2 ijerph-19-10432-f002:**
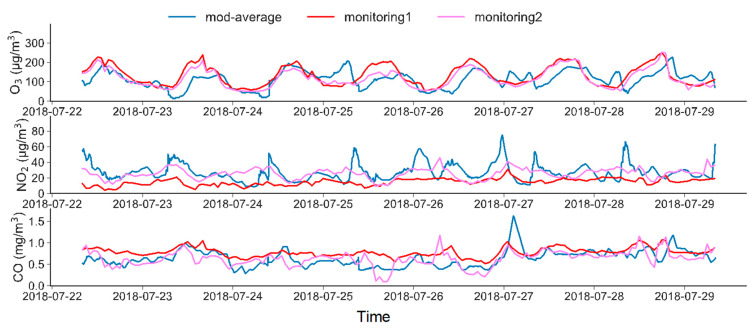
Time series of the observed and modelled O_3_, NO_2_ and CO concentrations from 22 July to 29 July 2018. The blue line (mod-average) represents the spatially averaged model data, while the red line (monitoring 1) and the purple line (monitoring 2) represent the observation data retrieved from monitoring sites 1 and 2, respectively.

**Figure 3 ijerph-19-10432-f003:**
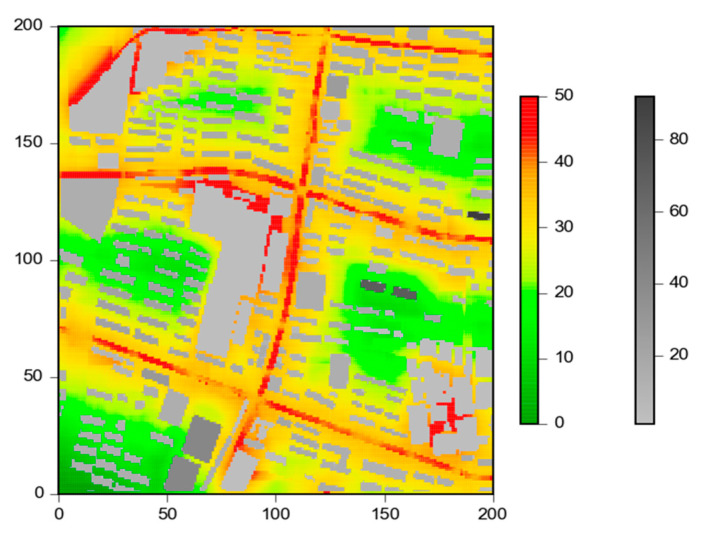
The distribution of the UMPI over the study area. Colors represent the values of the UMPI, and the grayscale represents building height.

**Figure 4 ijerph-19-10432-f004:**
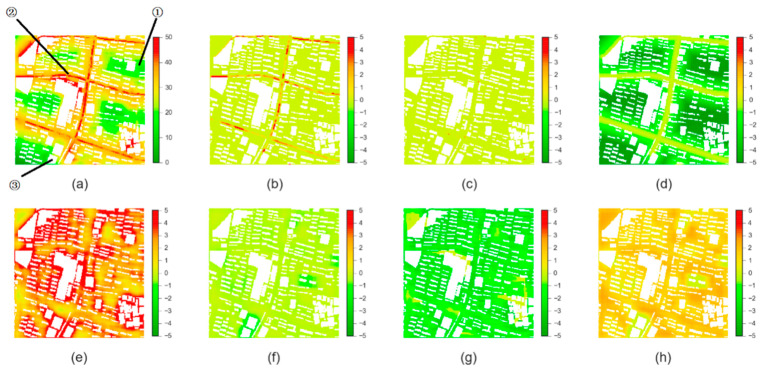
Contribution of different morphological indices to the UMPI. (**a**) the UMPI; (**b**) the Aspect Ratio; (**c**) the Asymmetry; (**d**) DR; (**e**) BCR; (**f**) the Rugosity; (**g**) the Occlusivity; (**h**) the Porosity.

**Table 1 ijerph-19-10432-t001:** Definition of the urban morphological parameters.

Parameter	Definition	Equation
BCR	Density of the building groups	$\mathrm{BCR}=\frac{S_{build}}{S_{total}}$
Rugosity	Average height of the urban canopy	$\mathrm{Rugosity}\;=\frac{\sum_{\mathrm{built}}A_{i}h_{i}}{\sum_{\mathrm{built}}A_{i}+\sum_{\mathrm{nonbuilt}}A_{j}}$
Porosity	Open area of pores	$\mathrm{Porosity}=\frac{\sum_{\mathrm{open}\;\mathrm{spaces}}V_{i}}{\sum_{open\;spaces}V_{i}+\sum_{built}V_{j}}$
Occlusivity	Openness of the urban spaces	$\mathrm{Occlusivity}=\frac{1}{N_{Horizcuts}}\underset{N_{Horizcuts}}{\sum }\frac{P_{built}}{P_{unbuilt}}$
Asymmetry ratio (*H*_1_/*H*_2_)	Asymmetry of the street canyon	$\mathrm{Asymmetric}\;\mathrm{ratio}=\frac{H_{1}}{H_{2}}$
Aspect ratio (*H*/*W*)	Depth of the street canyon	$\mathrm{Aspect}\;\mathrm{ratio}=\frac{H}{W}$
Distance from the main road (DR)	Shortest distance from the main road	

**Table 2 ijerph-19-10432-t002:** Multiple regression coefficients and coefficient of determination (R^2^) between the time-averaged concentrations of individual pollutants (i.e., NO_2_, O_3_, CO) and eight influencing factors at the ground level.

Pollutant	NO_2_ (µg·m^−3^)	O_3_ (µg·m^−3^)	CO (mg·m^−3^)
R^2^	0.71	0.47	0.77
Emission ^a,b^	0.56 *	−0.42 *	0.64 *
DR (m)	−0.28 *	0.25 *	−0.12 *
Aspect ratio (*H*/*W*)	0.06 *	0.02 *	0.06 *
Asymmetry (*H*_1_/*H*_2_)	0.13 *	−0.12 *	0.24 *
BCR	0.13 *	0.25 *	0.18 *
Rugosity	−0.03	0.03	−0.01
Occlusivity	−0.05 *	−0.03 *	−0.06 *
Porosity	0.05 *	0.03	0.05 *

^a^ The values in the table represent the standardized coefficients of the independent variables. ^b^ The emission value of NO_x_ is used for correlation with NO_2_ and O_3_. The emission value of NO_x_ is used for correlation with NO_2_ and O_3_. Data with the “*” symbol show significant correlation. To avoid multi hypothesis testing, the *p* value has passed the Bonferroni test (i.e., *p* < 0.05/8).

**Table 3 ijerph-19-10432-t003:** Multiple regression between the time-averaged concentration of individual pollutants (i.e., NO_2_, O_3_, CO) and urban form indices at the ground level.

Pollutant	NO_2_ (µg/m^3^)	O_3_ (µg/m^3^)	CO (mg/m^3^)
R^2^	0.53	0.35	0.53
DR (m) ^a^	−0.52 *	0.45 *	−0.41 *
Aspect ratio (*H*/*W*)	0.03 *	0.06 *	0.03 *
Asymmetry (*H*_1_/*H*_2_)	0.31 *	−0.27 *	0.44 *
BCR	0.06 *	−0.10 *	0.01
Rugosity	0.00	0.03	0.02
Occlusivity	−0.05 *	0.07 *	−0.03 *
Porosity	0.11 *	−0.11 *	0.09 *

^a^ The values in the table represent the standardized coefficients of the independent variables. Data with the “*”symbol show significant correlation. To avoid multi hypothesis testing, the *p* value has passed the Bonferroni test (i.e., *p* < 0.05/7).

**Table 4 ijerph-19-10432-t004:** Simple correlation coefficients between the standardized pollutants’ concentration (i.e., NO_2_, O_3_, CO) and individual standardized parameters over the road and non-road areas at the ground level.

Pollutant	NO_2_		O_3_		CO	
	Road	Non-Road	Road	Non-Road	Road	Non-Road
Emission	0.46 **	/	−0.48 **	/	0.57 **	/
DR	/	−0.15 **	/	0.22 **	/	−0.07 **
Aspect ratio (*H*/*W*)	0.24 **	/	−0.16 **	/	0.36 **	/
Asymmetry (*H*_1_/*H*_2_)	0.16 **	/	−0.13 **	/	0.23 **	/
BCR	0.01	0.04 **	0.05 **	−0.04 **	0.02	0.01 **
Rugosity	−0.27 **	−0.03 **	0.73 **	0.02 **	−0.15 **	−0.02 **
Occlusivity	−0.03 **	0.03 **	0.07 **	−0.03 **	−0.03 *	0.02 **
Porosity	0.09 **	0.01 **	−0.25 **	0.00	0.05 **	0.01 **

1. “/” indicates that this parameter is not applicable in that area. 2. *: *p* < 0.05; **: *p* < 0.01.

**Table 5 ijerph-19-10432-t005:** Correlation between the UMPI and the standardized pollution concentration (i.e., NO_2_, O_3_, CO) at the ground level.

	NO_2_	O_3_	CO
r	0.70	0.56	0.71
k	0.010	−0.008	0.010

## Data Availability

Data available on request due to privacy issues.

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
