# Peer review of "Evaluation of the Street Canyon Level Air Pollution Distribution Pattern in a Typical City Block in Baoding, China"

_ijerph, 2022, doi:10.3390/ijerph191610432_

Round 1

Reviewer 1 Report

I liked this manuscript dedicated to the evaluation of the street canyon level air pollution distribution pattern in Baoding, China. I could recommend this paper for publication after some minor editing.

 Lines 99-101. The author's assertion that the choice of this city is ideal for research is completely unconvincing. Authors should make statements other than the distance from Beijing.

The authors measured the parameters on the hottest days of the year July 22-29. How will the environmental situation change in the winter season?

Why was only the effect of transport pollution considered? It is necessary to indicate what proportion are the consequences of pollution from other sources of street pollution (heating, cooking, cleaning, etc.)

In my opinion, flows of at least one global pollutant should be described (for example, benzopyrene

In conclusion, the authors mentioned that their study has several limitations for use in other regions, I would like them to include a short outline of how their study will be carried out in the future.

Reviewer 2 Report

The authors investigated the street canyon level air pollution distribution pattern in a typical city block in Baoding, China. This study is well written, but a few concerns need to be taken into account.

1. Particulate matter is one of the most important air pollutants emitted from vehicle exhaust, but it was not considered in this study. What is the reason behind that?

2. The algebraic equation of multiple-linear regression used in the study should be incorporated in Section 3.2 in order to visualize what the unit of analysis for response and every single predictor variable is in this study.

3. Before presenting regression results, descriptive statistics for all variables presented in Tables 2 – 4 must be indicated and described in order for readers to understand the data, in which what types of data are in each regression model (continuous, discrete, or categorical data).

4. In Tables 2 – 4, standard error (SE) should be presented to indicate the uncertainty of the effect estimates.

5. The authors explored the association between urban morphological factors and concentration of different traffic air pollutants, including NO2, O3, and CO. How did authors justify whether or not there is multiple hypothesis testing (or multiple comparison problem) observed in this study.

6. The authors supplemented Figures S1 – S19 in supplementary material, but Figures S1, S8, S10 – S19 were not cited in the manuscript file, please either remove all aforementioned Figures from supplementary materials or cite all Figures in the manuscript file.

Round 2

Reviewer 2 Report

Thank you very much for addressing my concerns, and the manuscript is now getting better. I have no further comments.